# Google Earth Engine for Informal Settlement Mapping: A Random Forest Classification Using Spectral and Textural Information

**Dadirai Matarira** [1,*], **Onisimo Mutanga** [2] and **Maheshvari Naidu** [3]

1   School of Agriculture, Earth and Environmental Science, University of KwaZulu-Natal, P. Bag X01 Scottsville, Pietermaritzburg 3209, South Africa
2   Department of Geography, University of KwaZulu-Natal, P. Bag X01 Scottsville, Pietermaritzburg 3209, South Africa
3   Department of Humanities, School of Social Sciences, University of KwaZulu-Natal, Durban 4041, South Africa
*   Correspondence: dadimat19@gmail.com; Tel.: +27-629-915-107

**Abstract:** Accurate and reliable informal settlement maps are fundamental decision-making tools for planning, and for expediting informed management of cities. However, extraction of spatial information for informal settlements has remained a mammoth task due to the spatial heterogeneity of urban landscape components, requiring complex analytical processes. To date, the use of Google Earth Engine platform (GEE), with cloud computing prowess, provides unique opportunities to map informal settlements with precision and enhanced accuracy. This paper leverages cloud-based computing techniques within GEE to integrate spectral and textural features for accurate extraction of the location and spatial extent of informal settlements in Durban, South Africa. The paper aims to investigate the potential and advantages of GEE's innovative image processing techniques to precisely depict morphologically varied informal settlements. Seven data input models derived from Sentinel 2A bands, band-derived texture metrics, and spectral indices were investigated through a random forest supervised protocol. The main objective was to explore the value of different data input combinations in accurately mapping informal settlements. The results revealed that the classification based on spectral bands + textural information yielded the highest informal settlement identification accuracy (94% F-score). The addition of spectral indices decreased mapping accuracy. Our results confirm that the highest spatial accuracy is achieved with the 'textural features' model, which yielded the lowest root-mean-square log error (0.51) and mean absolute percent error (0.36). Our approach highlights the capability of GEE's complex integrative data processing capabilities in extracting morphological variations of informal settlements in rugged and heterogeneous urban landscapes, with reliable accuracy.

**Keywords:** cloud computing; heterogeneous urban landscapes; Sentinel 2A; textural features; data input combinations

## 1. Introduction

Informal settlements are a growing concern in urban landscapes, worldwide. In [1], informal settlements are described as overcrowded housing units that are constituted by fragile structures, often deprived of basic amenities such as safe water, sanitation, infrastructure and services, and lacking secure tenure. Recent statistics have indicated that, of the four billion people who are currently residing in urban areas [2], 1.6 billion live in informal areas [3], a figure that is estimated to rise to 3 billion by the mid-21st century [1]. According to the 2030 Agenda for Sustainable Development [4], countries are expected to increase efforts in upgrading and improving the quality of life of their residents. To support that vision, and guide upgrading processes [5,6], city planners and policy makers

need information on their location and extent [7], which is often scanty, not up-to-date, or inaccurate [6,8]. Furthermore, as informal settlements, particularly in Durban, South Africa, continue to be flood vulnerability hotspots [9] occupying precarious sites [10], determining their locations and extents provides baseline information for planning integrated management in the event of floods. Therefore, a realistic approach which allows production of consistent, reliable and comprehensive informal settlement morphologies is critical for disaster preparedness, and also as baseline data for supporting mitigation measures in the event of foreseen or unforeseen climate scenarios.

The formation, location and expansion of informal settlements are a result of multi-faceted and inter-related factors including, but not limited to, poor urban planning and management, uncontrolled population growth, rural-urban migration, inadequate housing provision [11,12] and the fact that in some sub-Saharan African countries, they are a manifestation of segregationist past [13,14]. Whilst economic globalization is associated with the spread of growth and greater opportunities [15,16], city authorities in developing countries fail to keep pace with increased urbanization in terms of provision of housing. Being characterized by erratic urban morphology [17], the emergence of informal settlements is reflective of increased inequalities and socio-economic disparities [15]. In the global north, some countries, such as the United Nations Economic Commission for Europe (UNECE) (18) countries, have also experienced these radical transformations.

To date, the increasing availability of remote sensing data has made whole city studies possible. Geospatial techniques have emerged as reliable tools for the capture of more detailed, accurate, up-to-date, and objective spatial information on informal settlements, their dynamics and their morphologic characteristics at high temporal frequency [5,18]. Traditionally, the measurement of informal settlements' extents was usually based on census data. However, because of the fluidity of informal settlements [19], survey-based information is often outdated [20], characterized by huge temporal gaps [21] and masks demographic and socio-economic differences in informal settlements [22]. Utilization of high-resolution imagery enables both spatial analytics and spectral analysis of informal settlements, [23], and are more efficient when compared with terrestrial surveys. However, obtaining reliable and accurate data on informal settlements continues to be hindered by (1) heterogeneity and complex spectral characteristics of urban land [24], (2) fragmented spatial configuration [25] and diversity of morphologies of informal settlements [26]. These characteristics vary extensively between countries, cities, within cities, and socio-economic contexts, making the characterization of spatial resolution and data input combinations difficult [27,28]. Given the fragmentation of urban landscapes, high spatial resolution often leads to high spectral mixing especially when spectral information is the sole data input [6]. For that reason, spatial contextual information in the form of image texture can be exploited in capturing their morphological variations [5,21,29,30].

Scientific research has been carried out in exploiting texture analysis to clearly identify and capture informal settlements [18,31]. In particular, several studies have explored texture feature algorithms such as grey level co-occurrence matrix (GLCM) [32–35], contourlets [36], curvelets [37], lacunarity [38–41], local Binary Patterns (LBPs) and Line Support Regions (LSRs) [42]. In other studies [18,42,43], spectral information, spectral indices and textural information have been integrated for improved accuracy, with NDVI being the widely used spectral index. For instance, [18] achieved accuracy levels of between 84% and 88% when grey level co-occurrence matrix (GLCM) variance was combined with NDVI, and an accuracy level of 90% when spectral information was combined with GLCM variance. However, such studies used limited numbers of input variables. Integration of image texture with spectral indices such as normalized difference water index (NDWI), the soil-adjusted vegetation index (SAVI) and the normalized difference building index (NDBI) for informal settlement detection to date, has rarely been exploited. Since urban areas are constituted by varied feature classes such as water bodies, built-up areas and vegetation [38], incorporation of the aforementioned spectral indices can help enhance class separability, thus contributing to increased informal settlement identification. Although the combination

of the band-derived features has the potential to enhance image classification accuracy, their computation is accomplished through the application of numerous, tedious and sometimes time-consuming functions. For example, the extraction of texture features, particularly grey level co-occurrence matrix (GLCM) texture features, is carried out through the application of numerous functions to the image bands at varied window sizes. This often results in huge volumes of input data [44]. There are also computation costs involved in averaging directions, as well as in texture feature selection. Apart from being time consuming [42], the handling of large datasets with many features usually results in computational limitations, especially for a personal computer, where classical image-processing software is concerned [45]. In addition, the integration of the various input parameters would generate high dimensional feature sets, resulting in a sheer volume of data processing that traditional image processing platforms may fail to handle, thus causing classification complexity [46,47].

Google Earth Engine (GEE), with its advancements in data processing and analytic tools, high computational power, and huge storage capacities [48], presents the potential to help overcome the limitations associated with handling voluminous data, in terms of storage, integration, processing, and analysis [28,49]. GEE's abundant imagery archives and data products [50], for example Landsat-8, Sentinel-1 and -2, and MODIS [51] mean that users do not need to download large datasets to local directories. Its integrative ability through effective script writing [52], and parallelized processing of a stack of images have offered opportunities for integrating different feature sets at great speed, making timely outputs a reality [53]. Furthermore, its provision of a complete package in terms of a plethora of remotely sensed images and cloud resources that warrant fast processing and analysis of images makes traditional software and desktop-based image analysis obsolete [54]. Given that background, several investigators have thus taken advantage of GEE cloud-computing for mapping purposes at diversified scales, ranging from global [55], continental [56,57], to country scale [28,58].

The application of GEE has been investigated in urban environments [52,59,60]. For instance, [47] integrated spectral, textural and topographical features in LULC in the Tigris–Euphrates basin. In another study, [47] utilized spectral indices and GLCM textural indices for object based LULC classification in Trasimeno Lake, in Umbria, Central Italy. In [27] Landsat 7 and Landsat 8 bands, vegetation indices, and GLCM textural features were used to obtain a land cover map in Mozambique. The researchers took advantage of an environment in GEE that allows building of composite images from integrated feature sets [61], summoning, processing, and stacking of image input data, running all analyses in parallel [62]. In one of the first studies on the application of GEE in informal settlement mapping, [63] took advantage of the aforementioned cloud computing capabilities of GEE to map informal settlements in Colombia, through integration of spectral bands and spectral band-derived indices. In Colombia, informal settlements mainly occupy steep escarpments and are along urban fringes [64]. The study did not explore the impact of adding image texture for informal settlement extraction. According to [65], texture features have the capability to quantitatively differentiate informal settlement morphological characteristics such as high densities, organic morphology and disarranged spatial patterns, from planned, organized and well-structured urban layouts.

The paper seeks to extend the work of [63] through integration of spectral bands, spectral indices and textural features for the precise mapping of informal settlements within the GEE, in a South African context. To the best of our knowledge, GEE cloud computing capabilities have not been investigated for the characterization of informal settlements in South Africa. Apart from that, [66] discovered that studies in South Africa [21,35,67–69] have mainly concentrated in Johannesburg city [35,67–69], Soweto township, and mostly used Quickbird imagery. Durban lacks coverage on application of image texture for the mapping of informal settlements. Durban is characterized by varied morphological patterns, ranging from lining traffic arteries, steep terrain (e.g., Bester's Camp (Inanda), open spaces, to being in proximity to river networks, for example, Palmiet River (e.g., Quarry

Road West). Texture analysis allows extraction of the diverse and explicit morphological features [29,42].

Owing to this background, the study seeks to test performance of various data input combinations for precise characterization of informal settlements through exploitation of GEE cloud computing capabilities in the heterogeneous landscape of Durban, South Africa. The paper presents an approach for the creation of a reproducible classification framework, which would allow for the production of consistent data on a regular basis.

Specifically, the objectives were to:

(1)　Present an operational framework based on various Sentinel 2A band-derived spectral and texture feature combinations for capturing informal settlements in Durban, South Africa.

(2)　Determine the extent to which GEE's data analysis capabilities can precisely depict morphologically diverse informal settlements in the Durban landscape.

(3)　Statistically assess the deviations in informal settlement spatial extents derived from comparison analysis between modelled outputs and reference area estimates.

The results exhibit a paradigm shift from classical image processing software and approaches for detection of informal settlements towards advanced cloud computing resources that simplify access to datasets and processing of large feature sets.

## 2. Materials and Methods

### 2.1. Study Area

The study area (Figure 1c) is located in the province of KwaZulu-Natal, South Africa (Figure 1a), and lies within Durban Metropolitan area (Figure 1b). Situated in the north-western part of Durban city, the study area extends to, approximately, 900 ha between longitudes 29.95°E and 29.98°E, and latitudes 29.8°S and 29.83°S. The study area forms part of the Umgeni catchment, lying to the south of the Umgeni River. It covers suburbs such as Clare Estate, Westville, and Reservoir hills. The area includes informal settlements such as Kennedy Road, Quarry Road, New Germany road, Palmiet zone 1, and Foreman Road. The topography of the area is steep and highly undulating, ranging from about 30 m to 120 m above sea level. Generally, Durban is characterized by a humid subtropical climate with a mean annual precipitation exceeding 1000 mm per annum [70]. It is also characterized by warm, wet summers and mild, dry winters. Most informal settlements in the study area are close to road networks such as Palmiet Road, Clare Road, Quarry Road, and New Germany Road. These roads follow a steep topography and often lead down to Umgeni River. The informal settlements' location on steep slopes, in proximity to road and river networks, coupled with characteristic fragile soils may contribute to their vulnerability to landslides and flood hazards during extreme climatic conditions. Most of the informal settlements are located within pockets of formal settlements or close to prominent institutions or business areas. For example, Reservoir Hills shopping center lies adjacent to the New Germany road informal settlement. Similarly, the University of KwaZulu-Natal, Westville campus, lies at an average distance of about 1.5 km from the three closest informal settlements. This is reflective of the morphology of informal settlements in Durban, where they capitalize on every inch of urban space in the city. The fact that about 75 percent of the metropolitan gross housing backlog of 305,000 units represent informal dwellings [71] shows high levels of accommodation crisis in Durban. Moreover, the housing units are constructed using corrugated iron; plastic, timber, and metal sheeting testify to the poverty and low income levels of the residents.

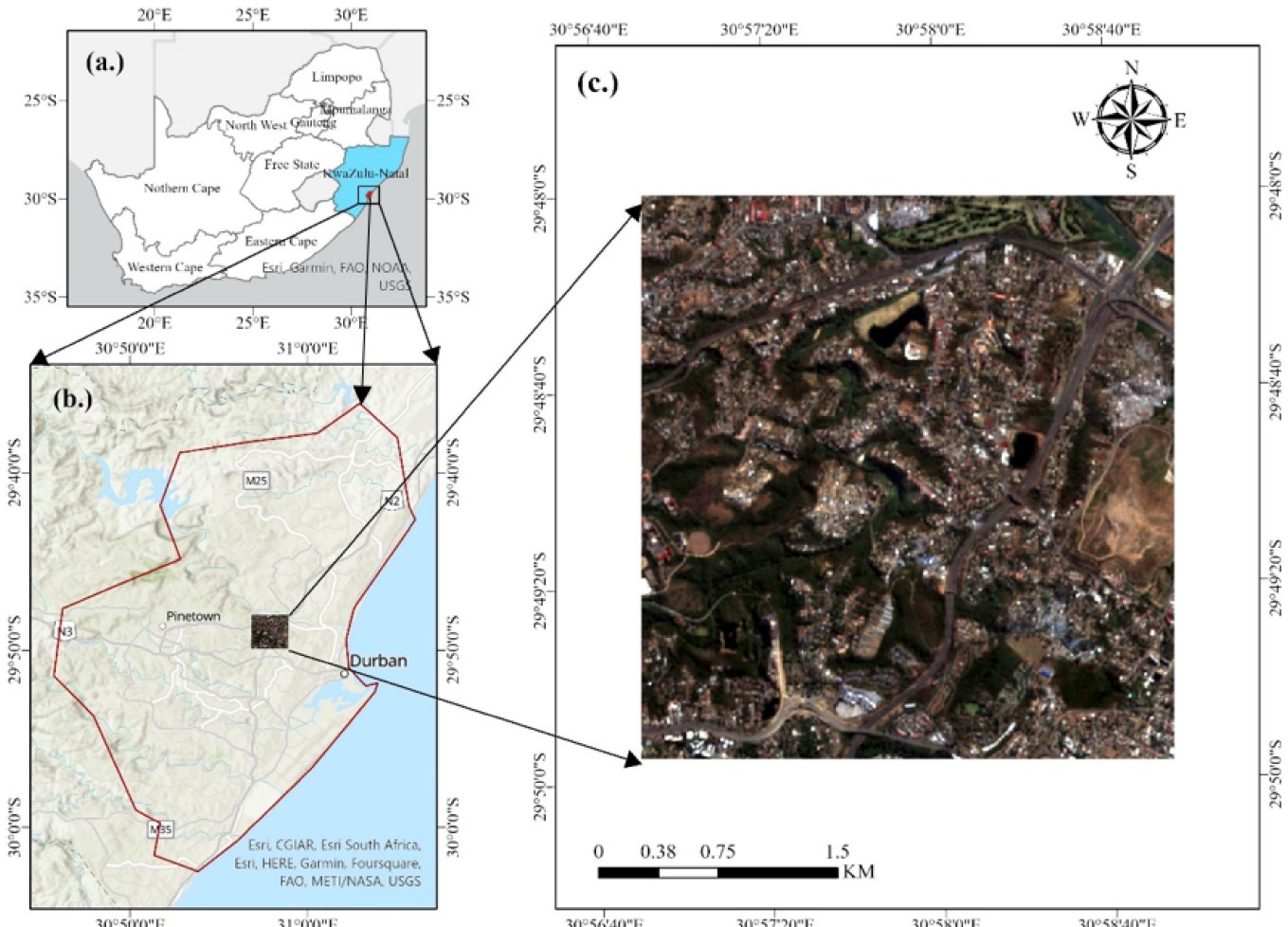

**Figure 1.** Study area selected in KwaZulu-Natal province (**a**), within Durban Metropolis (**b**), South Africa. (**c**) is the RGB overview of the Sentinel 2A imagery, in UTM/WGS84 plane coordinate.

### 2.2. Datasets

Sentinel 2A image collection (COPERNICUS/S2_SR surface reflectance dataset) was used in the analysis. The Sentinel 2A image covers 13 bands in the visible, near-infrared, and shortwave infrared (SWIR) wavelengths and consists of four bands at 10 m, six bands at 20 m, and three bands at 60 m [72,73]. To select data from the GEE archive, the filtered collection by date function was used. Multiple images covering a period from 1 August 2020 to 30 August 2020 in the study area were combined in a GEE collection. After filtering by date, 3 images were obtained that were used to form a composite, and a median value was assigned to each pixel. The resulting single image object represents the median value in each band of all the images in the filtered collection. Because clouds appear in different positions in the images, collections are a powerful way to removing many of the cloud-contaminated pixels [52,53]. Sentinel-2A image with less than 10% cloud coverage was employed.

### 2.3. Methods

Figure 2 illustrates the full approach adopted in this study. Our analysis consists of 7 methodological steps, which include: loading image collection and pre-processing, spectral feature extraction, texture feature calculation, feature input integration, feature importance evaluation, image classification, and accuracy assessment. GEE was the tool used to perform the bulk of the processing and analysis of Sentinel 2A imagery.

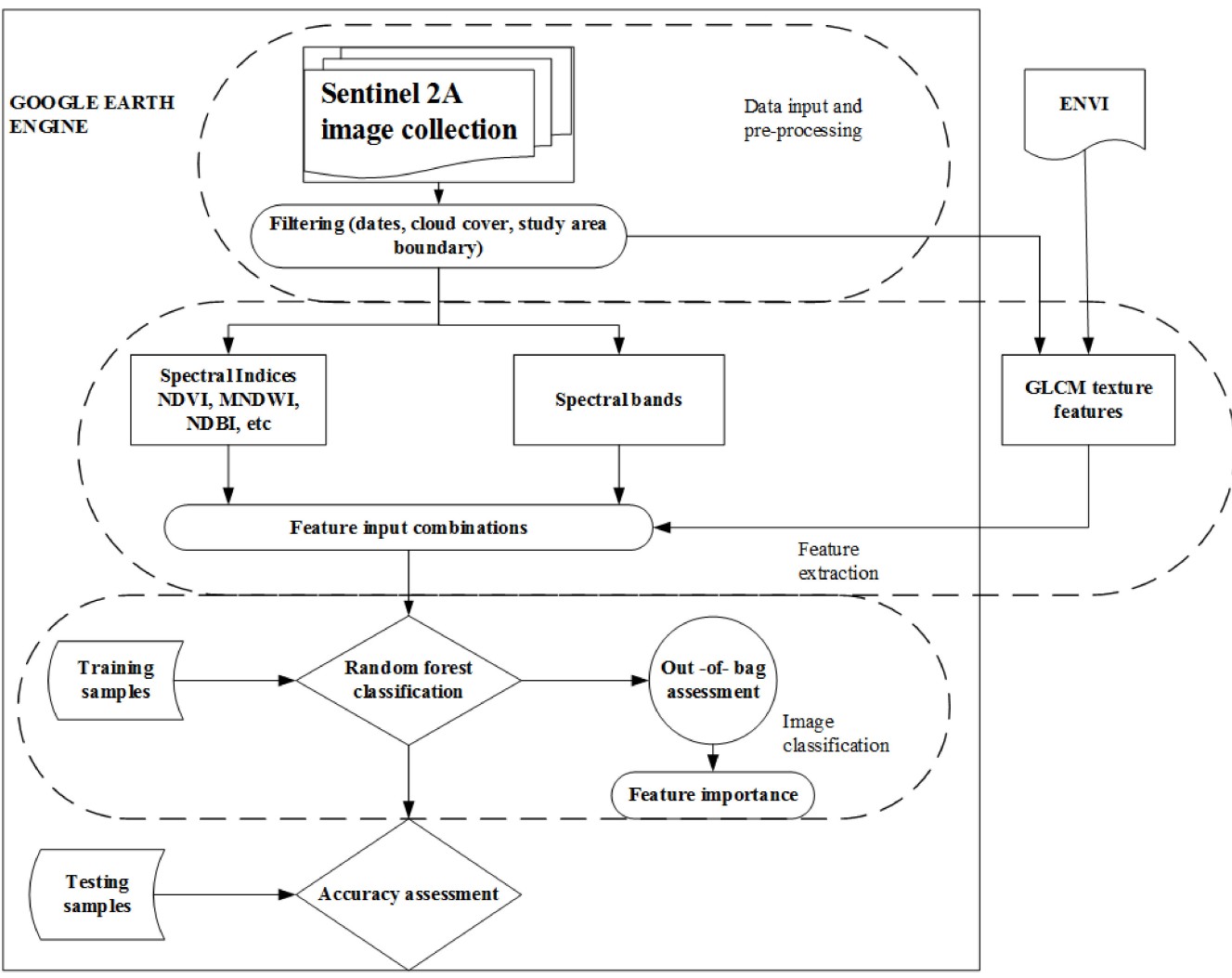

**Figure 2.** Research workflow chart.

2.3.1. Feature Extraction

In this study, both spectral and texture features were used as classification inputs. The extracted features included 10 spectral bands, 8 spectral indices, and 24 texture features, as shown in Table 1.

**Table 1.** Image feature sets that were extracted from Sentinel 2A imagery.

| Image Features | Names | Number of Features |
|---|---|---|
| Spectral bands (SBs) | Band (B2, B3, B4, B5, B6, B7, B8, B8A, B11, B12) | 10 |
| Spectral indices (SIs) | NDVI, NDWI, SAVI, NDBI, UI, NBI, BRBA, MNDWI | 8 |
| Texture metrics (Txts) | B2, B3, B4, (mean, variance, homogeneity, correlation, entropy, dissimilarity, contrast, angular second moment) | 24 |

Spectral Features

Following the reviewed literature [63,74], 10 spectral bands were selected as input features, which are B2, B3, B4, B5, B6, B7, B8, B8A, B11, B12 of Sentinel 2A imagery. Bands 1, 9, and 10 were not included because, according to [75], they are not meant for LULC classification. In addition to the spectral bands, eight spectral indices were

calculated from Sentinel 2A band combinations. These spectral indices were selected based on familiarity with the present land cover classes, such as urban, vegetation, and water [76]. Computed on the GEE cloud computing platform, these indices included two frequently used vegetation indices, namely NDVI and SAVI [28]. NDVI, being the widely used index in texture-based informal settlement detection [21,42,43], quantifies vegetation cover and better discriminates LULC classes. Calculation of SAVI involved the multiplication of the pixel values by a scale factor of 0.0001 to convert them to reflectance values, as recommended by [77].

Among the several spectral indices, water indices, namely, NDWI and the modified normalized difference water index (MNDWI) were included. Built-up area indices that were used in the current study included normalized difference building index (NDBI), urban index (UI), new built-up index (NBI), and the band ratio for built-up areas (BRBA). These indices have been previously incorporated for the extraction of built-up areas [78] and LULC mapping [79]. According to [74], the NDBI and UI provide fast detection of built-up areas or bare land. The mathematical equations used for the calculation of the aforementioned indices are presented in Table 2.

**Table 2.** Spectral indices selected for research.

| Spectral Index | Equation | Main Reference |
|---|---|---|
| NDVI | $\frac{B8-B4}{B8+B4}$ | [47,60,80] |
| SAVI | $1.5 \times ((B8 - B4)/(B8 + B4 + 0.5))$ | [77] |
| NDWI | $\frac{B3-B8}{B3+B8}$ | [47,80] |
| MNDWI | $\frac{B3-B11}{B3+B11}$ | [60,63] |
| BRBA | $\frac{B4}{B11}$ | [63] |
| NDBI | $\frac{B11-B8}{B11+B8}$ | [60,63] |
| NBI | $\frac{B4*B11}{B8A}$ | [60,63] |
| UI | $\frac{B7-B5}{B7+B5}$ | [63] |

GLCM Textural Features

To derive textural information, the GLCM algorithm was used. GLCM describes the probability of relationships between the reflectance values of neighbouring pixels at a distance and orientation invariant within the image [81]. The resultant raster layer that is made up of derived texture measurements may be input into the further analysis [82]. The metrics involved in the current study included angular second moment, contrast, correlation, variance, entropy, dissimilarity, mean, and homogeneity, computed from Sentinel 2A visible bands. According to [83], texture measurements can provide additional contextual information that enhances discrimination of diverse classes. The texture feature extraction was carried out using the GLCM implementation that is provided within the ENVI 5.3 software. In total, 24 texture features were obtained from the three visible bands. In their study on leaf area index estimation using textural features, [84] carried out the sensitivity analysis of the GLCM parameters and discovered that the most important parameters to be considered in image processing included orientation, displacement, and moving window size. Adopting that notion, GLCM texture measures were measured based on the average of all directions (0°, 45°, 90°, and 135°), the same co-occurrence shift (1,1), quantization level of 64, and 7 × 7 window size. According to [85], a quantization level of 64 preserves information and has an acceptable computing time. As pointed out in the literature [86–88], the window size is an important variable that has the potential to influence the discrimination capacity of extracted texture features. Selection of the optimal window size was performed using the method of coefficient of variation. The method of coefficient of variation was adopted from [86]'s study in their analysis of urban LULC classification. The process involved computation of class statistics for mean texture

feature, which, through visual analysis, showed the most discriminative capability for informal settlements. The class statistics included minimum, maximum, mean and standard deviation. These statistics were calculated for the red, blue and green bands and for the window sizes $3 \times 3$, $7 \times 7$, $9 \times 9$, $11 \times 11$, $13 \times 13$, $15 \times 15$. Coefficients of variation were calculated in Excel, using the formula CV = $\frac{\partial}{\mu}$, where

CV = coefficient of variation
$\partial$ = standard deviation
$\mu$ = mean

After texture feature extraction, the texture features were imported into GEE, exploiting the capability of the cloud computing platform to import and upload data on its public data catalog [89].

### 2.3.2. Feature Combinations

After the extraction of spectral and texture features from Sentinel 2A bands, combinations of various feature types were established. In that respect, 42 features were used to develop feature combinations that were incorporated in differentiating informal areas and other land uses. The input feature sets were composed of spectral bands (SBs), spectral indices (SIs), spectral bands plus spectral indices (SBS + SIs), texture metrics (Txts), spectral bands plus texture metrics (SBs + Txts), texture metrics plus spectral indices (SIs + Txts), and spectral bands + spectral indices + texture metrics (SBs + SIs + Txts) derived from sentinel 2A imagery. Based on the extracted features, 3 feature sets and 4 feature combinations were constructed to assess the influence of feature sets in distinguishing informal settlements.

### 3. Random Forest Classification

A pixel-based supervised RF machine learning algorithm was used for classification. RF classifiers applied to Sentinel 2A imagery in GEE have successfully mapped built-up areas [60], human settlements [90], and, specifically, informal settlements [63]. The choice of classifier for the current study was made owing to its capability to handle urban area classification where high-dimensional feature spaces are concerned and its robustness for informal settlement mapping in complex environments [91]. RF classifiers also measure each variable's contribution to the classification output, which is critical in assessing the value of each variable [61]. The entire classification process was performed in GEE where the building and tuning of the classification model were all based on the code "ee. Classifier package". The seven constructed feature sets were used as inputs in the classification models. Following [60], an RF model with 100 trees was created, and the number of variables per split was set to the square root of the number of variables [27,58]. Following [58], training samples were selected as small polygons to ensure that a polygon contains homogeneous pixels of a given land cover. In addition, small training polygons minimize the effect of spatial correlation [58]. The classification was completed using 782 training samples and 309 testing samples. The model was designed to perform a random sampling strategy to create approximately 70% of the training samples from the original dataset and generating a decision tree for each training sample separately, with the remaining 30% of the training samples being used as validation data for internal cross validation to evaluate the classification accuracy of the random forest [92]. Furthermore, five land cover classes that characterise the study area landscape were proposed in the land classification scheme. These included informal settlement, vegetation, water, formal buildings and bare land.

To obtain an accurate representation of the performance of the classifier [93], achieve better classifier stability [94], and consider potential variation in accuracy levels resulting from a random sampling of training samples [95], 20 replications of bootstrap sampling were performed and the same number of iterative classification trials performed for each model. RF algorithms apply a bagging approach involving randomly resampling training

data subsets to allow iterative construction of numerous, comparatively unbiased models which would then be averaged [95]. This means 20 classification results were obtained for each input feature set.

### 3.1. Variable Importance

In the current study, the RF algorithm was used for variable importance evaluation. In the evaluation of variable importance score, the value of a feature parameter is turned into a random number and the impact on the accuracy of the model calculated [92]. The importance of the parameter was calculated from the out-of-bag (OOB) error of each decision tree, calculated from the OOB data, following [94]. Feature importance evaluation was performed on feature combinations that consisted of more than 10 variables (Txts, SBs + SIs, SBs + Txts, SIs + Txts, and SBs + SIs + Txts) in order to obtain the 10 most significant variables, following [96,97]. Five new feature subsets (FSs), made up of the important variables from each evaluated model were used to train the classifier. After repetition of the classification process 20 times, average values of the 20 F-scores were calculated for each feature subset model. Performance evaluation of feature subsets was carried out through a comparison of classification results of different feature subsets against the original feature set from which they were derived. The assessment was completed in order to establish if feature reduction would significantly improve informal settlement identification or not.

### 3.2. Accuracy Assessment, Classification Comparison, and Statistical Testing

According to [61], accuracy assessment is critical in map production using remote sensing data. Establishing the relative comparative performances of different feature sets, against the SBs (benchmark experiment) was an important focus of this paper.

#### 3.2.1. Pixel-Based Accuracy Assessment

Classification performances of all seven classification models; SBs, SIs, Txts, SBs + SIs, SBs + Txts, SIs + Txts, and SBs + SIs + Txts were assessed. The confusion matrix implemented in GEE was used for the accuracy assessment of LULC classifications. User's accuracy (UA) and producer's accuracy (PA), calculated from the confusion matrix, were used to determine F-scores that were used as the accuracy metric for informal settlement identification. The calculated F-scores were representative of classification accuracies for the informal settlement class. F1-measures were used to quantify variations in the results for all feature-based models and the feature subset-based models. According to [80], the F1 score shows how good the classifier is in the context of both producer's and user's accuracies by weighting their average. The percentage deviations of F-score were calculated to assess the precision of the model vis-à-vis its accuracy.

$$\text{F} - \text{score} = 2 \ \times \ \frac{(UA)(PA)}{UA + PA} \tag{1}$$

F-score was calculated for each of the 20 classification iterations run for each feature set.

After performing 20 iterative classifications for a particular classification model, the 20 results of each experiment were tested for normality of distribution using the Shapiro test. Subsequently, the relative comparison between performances of pairs of different feature set combinations were performed using either the Welch Two-sample *t*-test or Wilcoxon Rank Sum test, depending on whether pairs of data were normally distributed or not. Where both datasets attained normal distribution, the Welch Two sample *t*-test was used to test if a significant difference existed between the means, with a corresponding calculation of *p*-values. Given the null hypothesis, $H_0$: $\mu_1 - \mu_2 = 0$ or $H_0$: $\mu_1 = \mu_2$ and alternative hypothesis H1: $\mu_1 - \mu_2 \neq 0$ or $H_0$: $\mu_1 \neq \mu_2$, $\mu_1$ and $\mu_2$ represented means for the classification results for models 1 and 2, respectively. The *p*-value was calculated for a 95% confidence level and the null hypothesis of equal means was rejected at $p < 0.05$. Where one dataset was normally distributed and the other one was not, the Wilcoxon Rank Sum test was carried out. Given the null hypothesis, $H_0$: $\eta_A = \eta_B$ and alternative

hypothesis $H_1$: $\eta_A \neq \eta_B$, $\eta_A$, and $\eta_B$ represented medians for the classification results for models A and B respectively. Similarly, the *p*-value was calculated at a 95% confidence level, and the null hypothesis of equal medians was rejected at $p < 0.05$. Particularly, the main aim of this analysis was to determine whether statistically significant differences existed between classification results of different feature input combinations. Statistical significance tests were also executed for each pair of predictions made by feature subset-based and all feature-based models.

### 3.2.2. Patch-Based Accuracy Assessment

For the patch-based accuracy assessment, seven informal settlements (labelled A-G) (Figure 3) were considered as independently derived reference data for spatial estimation of informal settlement areas. Boundaries for the respective informal settlements were digitized from Google Earth Pro and are shown in Figure 3. The polygons were used to compute areal estimates of informal settlements on the ground that were compared to the areal estimates on classified maps. The areas for the corresponding patches on classified maps were calculated using spatial analyst tools in ArcMap.

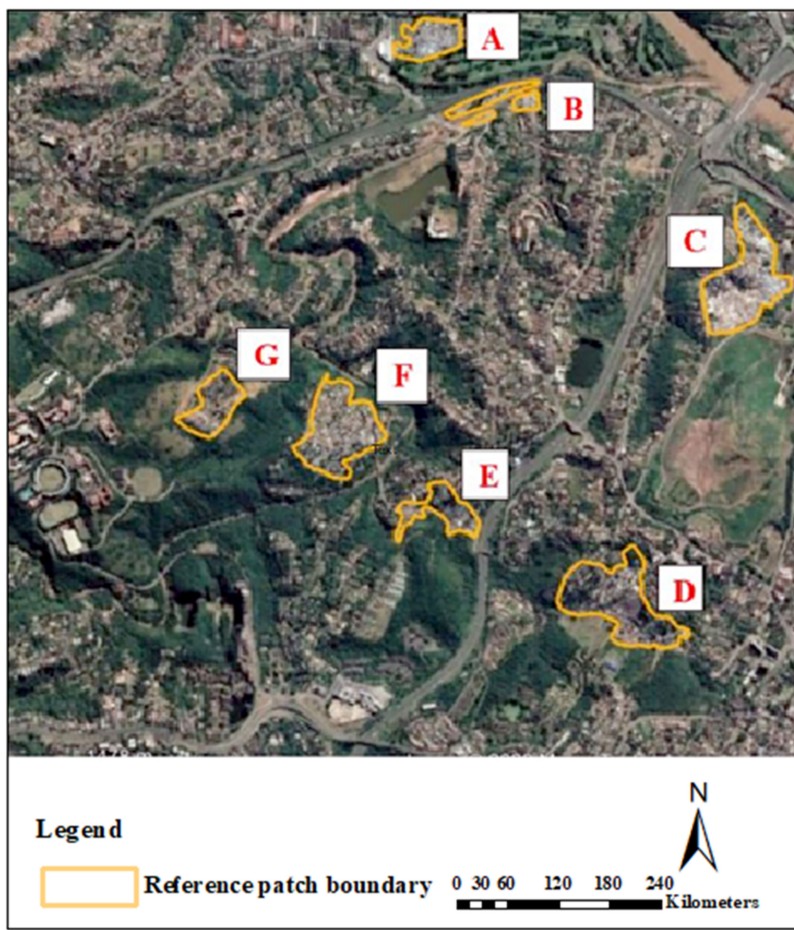

**Figure 3.** Ground truth samples of informal settlements (A–G (red)).

### Regression between Extracted Informal Settlement Areas and Ground Truth Data

Regression analysis was carried out to analyse the spatial variations in the relationship between ground truth area data and estimated areas. The focus of this analysis was to quantify the discrimination power of different feature input combinations through the measurement of the magnitude of error between predicted and observed spatial extents. In their research on spatial accuracy assessment of object boundaries, [98] suggested that validation concepts need to be extended to a spatial accuracy assessment of the objects' boundaries. In this regard, this paper presented two different spatial error assessment

methods, which included root-mean-square log error (RMSLE) and mean absolute percent error (MAPE). These error metrics, calculated in R statistical software, were used to compare the predictive ability of each input feature combination, in terms of deviation between boundaries of classification outputs and a reference dataset.

## 4. Results

### *4.1. Evaluation and Comparative Analysis of Classification Results*

4.1.1. Visual Analysis of Different Feature Input Models

The study tested seven different feature set options (1) spectral bands (SBs), (2) spectral indices (SIs), (3) texture features (Txts), (4) spectral bands and indices (SBs + SIs), (5) spectral bands and texture features (SBs + Txts), (6) spectral indices and texture features (SIs + Txts), and a combination of spectral bands, spectral indices and texture features (SBs + SIs + Txts). Through visual comparison, differences were noted in the classification outputs, paying particular attention to the degree of misclassification between the informal built-up land and other land uses. From a visual inspection, the drawback of "salt and pepper" effects could be evident in the models SIs, SBs + SIs, and SBs. Misclassifications were noticed, especially between informal areas and bare land. Some informal settlement patches could be seen in areas that, on the ground, were represented as bare land. Examples of misclassified areas are shown in black squares (Figure 4b,e,f). The black squares mark out areas with evident misclassifications of bare land and informal settlements. The distinct separation of informal settlements from other LULC classes could be identified in Txts, SBs + Txts, SBs + SIs + Txts, and SBs + Txts models where large and well-defined patches of informal settlements were labelled as informal built-up land.

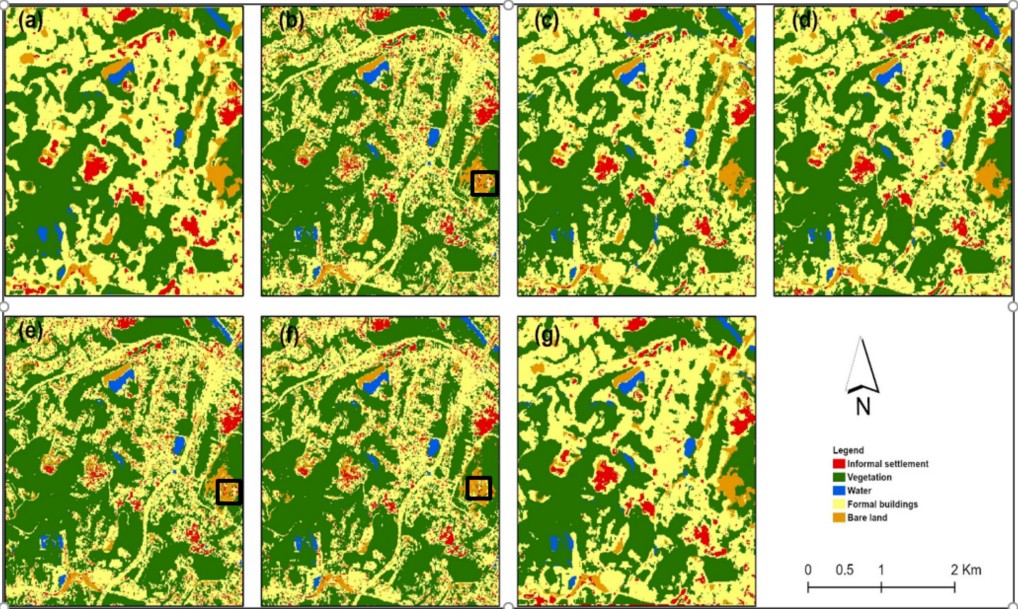

**Figure 4.** LULC maps obtained using different feature combinations from (**a**) Txts, (**b**) SIs, (**c**) SIs + Txts, (**d**) SBs + SIs + Txts (**e**) SBs + SIs, (**f**) SBs, (**g**) SBs + Txts.

4.1.2. Accuracy Assessment and Analysis

To assess the performance of the models, F-score, percentage deviation, and statistical tests were calculated for the various feature sets. Table 3 shows average F-scores, standard deviations, and statistical significance obtained by using SBs as a benchmark, and different types of features. The best accuracy results are displayed in bold.

**Table 3.** Producer accuracy (%), user accuracy (%), average F-scores and standard deviation for different feature input models.

| Model | PA | UA | F-Score |
|---|---|---|---|
| SBs | 83 | 89 | 86 ± 1.98 |
| SIs | 73 | 80 | 76 ± 2.06 |
| Txts | 86 | 94 | 90 ± 1.19 |
| SBs + SIs | 79 | 86 | 82 ± 1.43 |
| SBs + Txts | **91** | **97** | **94 ± 1.27** |
| SIs + Txts | 91 | 94 | 92 ± 0.91 |
| SBs + SIs + Txts | 91 | 96 | 93 ± 1.08 |

　　　Generally, accuracy was high for all feature-based models, averaging beyond 80%. The average F-scores ranged from 94 ± 1.27% (SBs + Txts) to 76 ± 2.06% (SIs). In fact, the descending order of IS identification accuracy was 94% (SBs + Txts), 93% (SBs + SIs + Txts), 92% (SIs + Txts), 90% (Txts), 86% (SBs), 82% (SBs + SIs), and 76% (SIs). Generally, the results suggested that the inclusion of image texture significantly boosted classification accuracy, since classification performances were enhanced in all the models that incorporated textural features. More specifically, the addition of textural features to spectral bands yielded the highest accuracy improvement when spectral bands alone were used as a benchmark experiment. On the other hand, combining spectral bands and spectral indices decreased the F-score by 4%. The results also show that the sole use of spectral bands resulted in relatively higher performance than "spectral bands+spectral indices". Considering models that incorporated textural features, the descending order of importance was SBs + Txts (94%), SBs + SIs + Txts (93%), SIs + Txts (92%), and Txts (90%). It is also important to note that, whilst the addition of textural information to spectral bands (SBs + Txts) yielded the highest accuracy levels, results declined when spectral indices were added to the feature set. Almost similarly, compared with Txts alone, the composite of spectral indices and texture metric (SIs + Txts) yielded decreased accuracy. Result analysis also demonstrated that combining multispectral bands with texture features performed better (94%) than using each type of feature solely (SBs-86%, Txts-90%). Furthermore, a composite of all three feature sets (SBs + SIs + Txts) did not provide superior results. Most importantly, all six experiments that were compared against the performance of only spectral bands showed significant differences between classification performances ($p < 0.05$) (Table 4).

**Table 4.** Two sample *t*-tests for the mapping of informal settlements and their *p* values.

| First Model | Second Model | Mean of First Model | Mean of Second Model | *p*-Value |
|---|---|---|---|---|
| **SBs vs** | SIs | 86 | 76 | < 0.05 |
| **SBs vs** | Txts | 86 | 90 | <0.05 |
| **SBs vs** | SBs + SIs | 86 | 82 | <0.05 |
| **SBs vs** | SBs + Txts | 86 | 94 | <0.05 |
| **SBs vs** | SIs + Txts | 86 | 92 | <0.05 |
| **SBs vs** | SBs + SIs + Txts | 86 | 93 | <0.05 |

　　　Whilst the accuracy deviations, as represented by standard deviations, were not very large, there was no consistency in terms of precision. For instance, considering the 20 test runs for the SBs + Txts models, its accuracy was the best, but its precision, as measured by standard deviation, was not. Although SIs + Txts model yielded lower accuracy than SBs + Txts, the result showed more precision as indicated by the low standard deviation. The percentage deviations of average F1-scores were 1.98%, 2.06%, 1.19%, 1.43%, 1.27%, 0.91%, 1.08% (Table 4). These standard deviations were for SBs, SIs, Txts, SBs + SIs, SBs + Txts, SIs + Txts, and SBs + SIs + Txts, respectively.

### 4.2. Importance of Features for Informal Settlement Mapping

Figure 5 shows the 10 most important variables of the models Txts, SBs + SIs, SBs + Txts, SIs + Txts, and SBs + SIs + Txts. The results demonstrated that there was no consistency in terms of performance of feature subset-based models. Whilst some feature–subset models FS (SBs + Txts), FS (SIs + Txts), and FS (SBs + SIs + Txts) achieved significant decreases in classification accuracy (Table 5), feature reduction did not yield any changes in classification accuracy for models such as Txts and SBs + SIs. In addition, features with high importance scores changed with changes in feature combinations. For instance, variable importance scores indicated that $B2_{mean}$, $B3_{mean}$, $B4_{mean}$, and $B4_{variance}$ were the four most important variables in the Txts model. For the SBs + Txts model, $B2_{mean}$, $B3_{mean}$, $B4_{mean}$, and B8A were the most important. In the SIs + Txts model, $B2_{mean}$, $B3_{mean}$, $B4_{mean}$, and *ui* contributed the most information to the model. Similarly, $B2_{mean}$, $B3_{mean}$, $B4_{mean}$, and *ui* were the most important variables in the SBs + SIs + Txts model. Considering SBs + SIs model, B2, B12, *ui*, and ndwi had the most influence in the classification.

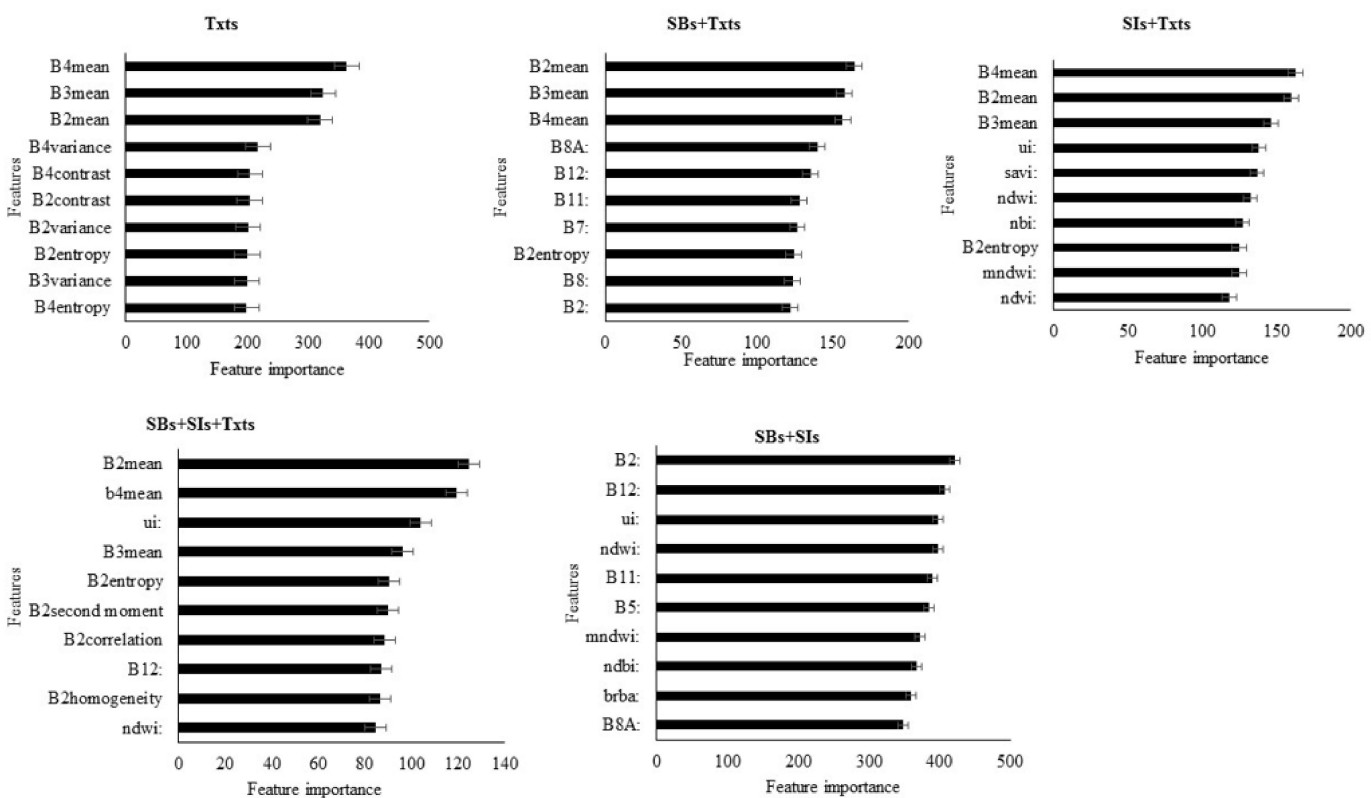

**Figure 5.** Feature importance scores of the 10 most important features for the image combinations.

**Table 5.** Two sample *t*-tests for all variables vs feature subsets for feature input models.

| Feature Input Model | F-Score | | *p*-Value |
| | All Variables | Feature Subset | |
| --- | --- | --- | --- |
| Txts | 90 | 90 | |
| SBs + SIs | 82 | 82 | |
| SBs + Txts | 94 | 90 | $p < 0.05$ |
| SBs + SIs + Txts | 93 | 88 | $p < 0.05$ |
| SIs + Txts | 92 | 84 | $p < 0.05$ |

In the feature combinations where spectral indices were incorporated, *ui* obtained the highest importance score and would feature in the top four variables. When textural features were incorporated, $B2_{mean}$, $B3_{mean}$, and $B4_{mean}$ attained the highest importance score, indicating the relevance of mean texture in the classification process. There was no

consistency with regard to the most important spectral band, with B8A contributing most in the model SBs + Txts and B2 contributing most in the SBs+ SIs model. From Figure 5, it could be observed that the variable importance score also decreased when the number of features used to build the RF model increased. For example, considering the Txts model, $B2_{mean}$, $B3_{mean}$, and $B4_{mean}$ attained feature scores ranging between 300–400. When Txts were combined with either SBs or SIs, importance scores were reduced to between 150 and 200. The feature importance score was reduced further to between 100 and 140 when Txts was combined with both SBs and SIs.

Feature Subset Evaluation

From the two-sample *t*-tests that were carried out, three models showed significant differences between the performance of feature models and feature subset-based models. Classification results for feature subsets extracted from SBs + Txts, SBs + Txts + SIs, and Txts + SIs models were significantly lower than that for their corresponding all feature models. This suggests that feature reduction caused a decline in classification results. The average decrease in classification was 6%. On the other hand, there were no observed differences in informal settlement identification accuracy for the subsets derived from Txts and SBs + SIs. Table 5 shows the performance of feature subsets against the feature sets from which the subsets were selected. The same trends of accuracy measures were observed for the feature-subset-based models as for the feature-based models, where the descending order of F1 measures was FS (SBs + Txts), FS (SBs + SIs + Txts), and FS (SIs + Txts).

### 4.3. Patch-Based Accuracy Assessment

The results demonstrated the underestimation of informal settlement extents by all the models. Figure 6 shows informal settlement patches overlaid on digitized polygons for the same settlement. The results indicate that the informal settlement patches on classified maps covered smaller areas than the real areas on the ground. Calculations of relative spatial errors demonstrated high RMSE values for all the models ranging from 0.51 to 1.2 and MAPE values ranging from 0.36 to 0.61. For both RMSLE and MAPE, the results in Table 6 indicate that the Txts model yielded the best spatial accuracy (RMSLE = 0.51; MAPE = 0.36). Table 6 presents a summary of classified and ground truth areas for selected informal settlements.

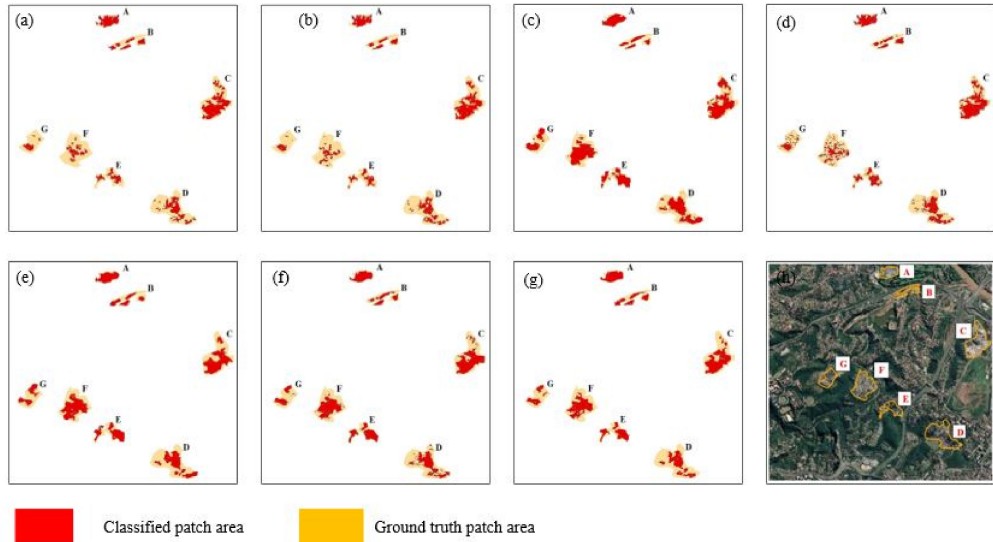

**Figure 6.** Classified informal settlement patch areas (A–G (black)) for models (**a**) SBs, (**b**) SIs, (**c**)Txts, (**d**) SBs + SIs, (**e**) SBs + Txts, (**f**) SIs + Txts, (**g**) SBs + SIs + Txts, and (**h**) corresponding ground truth polygons (A–G (red)).

**Table 6.** Spatial accuracy assessment results for area-based classification.

| Classification Model | Patch | Classified Patch Area (ha) | Reference Patch Area (ha) | Difference | Difference (%) | RMSLE | MAPE |
|---|---|---|---|---|---|---|---|
| SBs | A | 2.97 | 3.94 | 0.97 | 24.62 | | |
| | B | 1.89 | 1.86 | −0.03 | −1.61 | | |
| | C | 6.96 | 12.43 | 5.47 | 44.01 | | |
| | D | 3.89 | 13.55 | 9.66 | 71.29 | 1.13 | 0.57 |
| | E | 1.83 | 4.49 | 2.66 | 59.24 | | |
| | F | 1.97 | 11.22 | 9.25 | 82.44 | | |
| | G | 0.95 | 5.09 | 4.14 | 81.34 | | |
| SIs | A | 2.72 | 3.94 | 1.22 | 30.96 | | |
| | B | 1.50 | 1.86 | 0.36 | 19.35 | | |
| | C | 6.76 | 12.43 | 5.67 | 45.62 | | |
| | D | 3.80 | 13.55 | 9.75 | 71.96 | 1.2 | 0.61 |
| | E | 1.88 | 4.49 | 2.61 | 58.13 | | |
| | F | 1.82 | 11.22 | 9.4 | 83.78 | | |
| | G | 0.83 | 5.09 | 4.26 | 83.69 | | |
| Txts | A | 3.25 | 3.94 | 0.69 | 17.51 | | |
| | B | 1.73 | 1.86 | 0.13 | 6.99 | | |
| | C | 7.83 | 12.43 | 4.6 | 37.01 | | |
| | D | 6.47 | 13.55 | 7.08 | 52.25 | 0.51 | 0.36 |
| | E | 3.85 | 4.49 | 0.64 | 14.25 | | |
| | F | 6.70 | 11,22 | 4.52 | 40.29 | | |
| | G | 2.62 | 5,09 | 2.47 | 48.53 | | |
| SBs + SIs | A | 3.11 | 3.94 | 0.83 | 21.07 | | |
| | B | 1.80 | 1.86 | 0.06 | 3.23 | | |
| | C | 7.83 | 12.43 | 4.6 | 37.01 | | |
| | D | 4.97 | 13.55 | 8.58 | 63.32 | 0.88 | 0.50 |
| | E | 2.61 | 4.49 | 1.88 | 41.87 | | |
| | F | 3.04 | 11.22 | 8.18 | 72.91 | | |
| | G | 1.28 | 5.09 | 3.81 | 74.85 | | |
| SBs + Txts | A | 3.88 | 3.94 | 0.06 | 1.52 | | |
| | B | 1.42 | 1.86 | 0.44 | 23.66 | | |
| | C | 8.31 | 12.43 | 4.12 | 33.15 | | |
| | D | 5.28 | 13.55 | 8.27 | 61.03 | 0.63 | 0.38 |
| | E | 3.63 | 4.49 | 0.86 | 19.15 | | |
| | F | 6.44 | 11.22 | 4.78 | 42.60 | | |
| | G | 2.12 | 5.09 | 2.97 | 58.35 | | |
| SIs + Txts | A | 2.93 | 3.94 | 1.01 | 25.63 | | |
| | B | 1.85 | 1.86 | 0.01 | 0.54 | | |
| | C | 7.04 | 12.43 | 5.39 | 43.36 | | |
| | D | 5.04 | 13.55 | 8.51 | 62.80 | 0.68 | 0.44 |
| | E | 3.53 | 4.49 | 0.96 | 21.38 | | |
| | F | 5.71 | 11.22 | 5.51 | 49.11 | | |
| | G | 1.82 | 5.09 | 3.27 | 64.24 | | |
| SBs + SIs + Txts | A | 3.11 | 3.94 | 0.83 | 21.07 | | |
| | B | 1.70 | 1.86 | 0.16 | 8.60 | | |
| | C | 6.75 | 12.43 | 5.68 | 45.70 | | |
| | D | 4.45 | 13.55 | 9.1 | 67.16 | 0.73 | 0.46 |
| | E | 3.19 | 4.49 | 1.3 | 28.95 | | |
| | F | 5.50 | 11.22 | 5.72 | 50.98 | | |
| | G | 1.83 | 5.09 | 3.26 | 64.05 | | |

## 5. Discussion

The study sought to investigate performance of different feature input combinations in accurately depicting morphologically varied informal settlements as well as their spatial

extents within the GEE platform. Variable results were obtained depending on the input datasets. Average accuracy of >80% suggests the success of RF in extracting informal built-up areas in the study area. Results demonstrated that classification of spectral bands alone yielded relatively low model performance (86%), whilst models that incorporated texture features performed better, ranging from 90% to 94%. More specifically, the combination of spectral bands and textural features yielded the highest accuracy of 94%. Comparably, in their study, [18] achieved an increase in accuracy from 62% to 65% when image texture was integrated with spectral bands. These results are consistent with the current results where accuracy increased from 86% (spectral bands alone) to 94% (spectral bands + texture features). Both results are supported by [47], who alluded that spectral bands alone are insufficient in discriminating different LULC types, and that similar morphological characteristics, in the form of constructional materials, paint or roof colours [99,100] can help explain confusion within urban landscapes. GLCM texture-based analysis can capture urban morphological characteristics such as built up densities [31], shape, size, orientation and roof colours [29]. However comparing the classification performances, RF classification within the GEE platform, performed in the current study, yielded significantly higher accuracies than RF classification implemented by [18] in the eCognition software. Given some similarities in morphological characteristics of informal settlements in Mumbai and Durban (high densities, organic morphology, iron and asbestos roofing sheets), higher accuracy in the current study can largely be explained in terms of GEE's integrative ability through effective script writing [52], and parallelized processing of a stack of input features, that offers opportunities for integrating different feature sets for enhanced mapping accuracy. Whilst 24 texture variables were integrated in the current study, [18] only used variance. In [42], it is argued that combining a number of texture descriptors is crucial in discriminating complex urban settlement patterns. Mirroring these findings, RF models within the GEE were able to capture the morphological characteristics of informal settlements better than studies that used classical software.

Results also indicate that, while the addition of spectral indices to the "spectral bands + texture features" model significantly reduced the accuracy level from 94% to 93% in the current study, the addition of NDVI to the "spectral bands + texture" model increased informal settlement identification accuracy in [18]'s study, from 65% to 90%. The results from [18] showed the relevance of NDVI in distinguishing informal settlements in Mumbai. Conversely, considering combinations that incorporated spectral indices, variable importance analyses indicated that NDVI only featured in 1 out of 3 feature subsets (Figure 5), where it assumed the lowest rank. The low performance of NDVI in the current study could also help explain the reduced accuracies when spectral indices were incorporated into other feature sets. Nonetheless, the current results appear consistent with [42]'s findings, where NDVI resulted in consistently low accuracies. This result is potentially explained by the lack of variance in urban vegetation in many cities [42,51]. In addition, the significant fall in classification accuracy in the current results when spectral indices were added to the "spectral bands + texture features" model is in agreement with other studies in spatially heterogeneous complex landscapes [101,102]. In their mapping of complex surface-mined and agricultural landscapes, [101] attributed reduced accuracy to the importation of redundant information, since spectral indices are derived from the linear computation of spectral bands [103]. Furthermore, [104] argued that the main problem with spectral indices being used for urban area mapping is that they cause spectral mixing between built-up areas and bare surfaces due to their similarity in spectral response patterns, especially in spatially heterogeneous environments. These explanations conform with the current results where misclassifications of bare land as informal settlements were evident (see the black rectangles in Figure 4b,e) in models that incorporated spectral indices (Figure 4). In their study, [80] also observed similar spectral behaviours between built-up and bare lands. Most importantly, the current results mean that, although informal settlement mapping can be performed better through the RF algorithm, some problems of spectral confusion, especially between bare surfaces and built-up areas, remain unsolved.

Examination of the performance of feature subset-based models revealed that feature subsets either yielded significantly reduced classification accuracies ($p < 0.05$) at a 95% confidence interval, with an average value of 6%, or yielded no change in classification results. This finding agrees with [42]'s findings in which feature reduction caused decreased accuracy, and no feature subsets were as powerful as the full combination. The current findings are, however, inconsistent with [105], who in their mapping of a complex mining environment, reported significant improvement in accuracy from the utilization of the top 10% of variables selected using variable importance measures. The fact that combinations of different input features yielded higher results than single feature sets suggest that, although adding additional features such as textural variables to the original spectral bands increases the dimensionality of feature space [42], more input variables could enhance informal settlement discriminability. This observation is consistent with [51], who alluded that extraction of many features from satellite data is one way of increasing image classification accuracy. Current findings contribute to revealing the strength of the RF classifier in dealing with high-dimensional feature sets [83].

*Estimated Informal Settlement Areas*

From the analysis of modelled slum patches and ground truth samples of slum patches, the results indicate that, although distinct informal settlement patches and clear boundaries could be identified from RF classification, there were inconsistencies in the mapping. Results demonstrated evidence of underestimation of informal settlement spatial extents (Figure 6). Generally, high RMSLE values were evident ranging from 0.51 to 1.2. Whilst mapping results suggest the effectiveness of RF classification in capturing informal settlement locations, underestimation indicates lack of robustness in the capturing of their diversity. The inconsistences could be attributed to the complexity of informal settlement morphologies [106]. Challenges exist in capturing informal settlements characterized by varied typologies within the area. For instance, remote sensing data fail to reveal the dynamics in factors that shape the apparent morphologies of the informal settlements, for example, culture, socio-political and economic status. Capturing informal settlements at varied stages of development could be challenging [43]. Whilst remotely sensed data and socio-economic parameters of an area may correlate [65], structural variations emanating from a high socio-economic gradient between two informal settlement areas could compromise the reliability of results. In [29], it is suggested that integrating remotely sensed data with survey-derived socio-economic information such as employment status, educational status, population figures and population density would avoid representation of informal settlements as one-dimensional.

In the current study, there were also some limitations. Firstly, the near-infra-red band which was not used in the extraction of image texture could be exploited, since according to [83], it provides greater contrast and thus carries the most significant data spikes. Secondly, our method utilized Sentinel 2A images with relatively low spatial resolution, which could explain some false identification that was evident in the results. The definition of adequate spatial resolution has been regarded as a key issue in the mapping of complex environments [27,107]. For instance, the potential of PlanetScope data for precise mapping of heterogeneous landscapes, for example, in identifying crop types and extents in small holder environments, has been emphasized [108,109]. In addition, the application of object-oriented analysis techniques, particularly within GEE, is constantly evolving [52]. In [110], land cover was mapped using object-based image classification and PlanetScope imagery within GEE and increased accuracy was achieved. Further, the author integrated PlanetScope with Sentinel 1 and Sentinel 2 data using the object-based oriented approach and achieved improved geometric and thematic accuracy. These findings present an opportunity to explore the capabilities of high resolution PlanetScope and object-oriented analysis in overcoming challenges of inaccurate identification of dynamic, spatially and morphologically complex informal settlements. Accurate and consistent characterization

of informal settlements would provide insights into their historical and contemporary dynamics, as well as in simulating future land changes.

## 6. Conclusions

GEE cloud computing was successfully applied for informal settlement mapping in part of Durban Metro, South Africa. GEE showed considerable versatility and adaptability due to its integrative capabilities and its efficient platform for script writing. Within the GEE environment, this work developed and tested pixel-based classification of various input combinations. The best performing input variables for the random forest ensemble classifier were identified through systematic testing of different feature input combinations. The RF model performed well in distinguishing informal settlements, yielding an average accuracy above 80%. The addition of texture features yielded statistically significant accuracy levels, whilst the addition of spectral indices generated significantly reduced accuracy levels. Considering the accuracy level of the informal settlement class, the spectral bands + texture features model achieved the best performance (94%). The texture features model yielded the lowest spatial error, enabling it to most accurately depict informal settlement boundaries.

The results demonstrate how the GEE framework, by simplifying access and processing of a large amount of satellite data, is shifting the paradigm in built-up area mapping from a static, product-based approach into a more dynamic and application-specific one with reasonable accuracy and in no time.

**Author Contributions:** D.M. developed the aim and objective of the research, including conceptualization of manuscript, data and result analysis, as well as the write-up of the initial draft manuscript. O.M., as the main supervisor, verified the analytical approaches and results of the study, including interpretation and discussion of the obtained results. He also gave suggestions towards refining of the novelty of the manuscript and coherence of the entire manuscript, starting from the introduction to the discussion. M.N., as my co-supervisor, also gave constructive criticism and comments which improved our manuscript. All authors have read and agreed to the published version of the manuscript.

**Funding:** This research did not receive any specific grant from funding agencies in the public, commercial, or not-for-profit sectors.

**Acknowledgments:** This study was supported by the UKZN Big Data for Science and Society Project (BDSS).

**Conflicts of Interest:** The authors declare no conflict of interest.

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
