# Peer review of "Google Earth Engine for Informal Settlement Mapping: A Random Forest Classification Using Spectral and Textural Information"

_remotesensing, doi:10.3390/rs14205130_

Round 1
Reviewer 1 Report
See attached file for comments and minor corrections

Reviewer 2 Report
The manuscript poses the classification problem for informal settlement mapping and sued the Sentinel-2 and GEE platform for this task. While the manuscript presents the problem and potential gaps, the innovation aspects are not well justified, and resulted mapping does not provide a critical advance or solution for the new operational products as well (see the quality of results in fig. 4 and 7). The GEE application can benefit a large area mapping in multiple locations but this study uses a single location for this task. It should prove the application in many other cities to maximize the contribution of this analysis because Random forest and feature selection are well-known standard procedures for mapping, and the authors should not emphasize it as an innovation aspect. The critical problem for informal settlement mapping is remote sensing spatial resolution. 20-m Sentinel-2 is not adequate for high accuracy area mapping of small and irregular residential building areas, and commercial datasets should be explored. This is a challenge for any application on urban context because the dataset is not freely available. Moreover, the manuscript can be improved in many ways, especially the introduction and discussion sections. For these reasons, I recommend a major review of this manuscript, and critical advances are expected to consider this manuscript for publication.
Major comment 1:
Introduction and Discussion sections are very long and they should be reduced for better readability. This is critical for manuscript approval.
Major comment 2:
Specific objectives highlight the GEE for LULC maps and feature selection. Both objectives are very common in previous papers, and informal settlement mapping is the only innovative aspect of this manuscript. Can you emphasize other factors that make this manuscript publishable?
Comment 1:
Paragraph 50 – 57. The authors should adjust this paragraph because remote sensing does not deliver the detailed level of information compared to Census. Remote sensing allows mapping of settlement (area, perimeter, complexity, building size/counts) but it does not substitute the Census variables (income, family size, education level), and this paragraph leads to wrong insight that remote sensing can replace it. I believe it is good content but should be slightly adjusted to express the “complementary information” that remote sensing provides for urban planners.
Comment 2:
While there is no consensus, I assume that very high resolution is below 5-m and Sentinel-2 is not VHR imagery. Consider adjusting it.
Comment 3:
Line 194 – 197: This statement is not strong to justify the study because we know that feature selection is specific for each context, and many studies are adopting certain feature selections. Beyond that, what is the novel aspect of this study?
Comment 4:
Line 199: What is high accuracy? Overall accuracy 70% or 99%? I suggest the rephrase of this part.
Comment 5
Section 2.2. Datasets: Why did you select August 2020? How many images were used in this composite?
Comment 6:
Table 3 is not informative and it seems to be a figure. You should improve it.
Comment 7:
Do you think that settlements dataset with seven polygons is sufficient for accuracy assessment?
Comment 8:
Figure 6 shows only validation on seven settlements but the mapping result in fig 7. shows many noise areas. What are your criteria to only validate seven settlements and not include the other areas in Fig. 7?
Line 47: Change from “to do” to “related to”
Line 50: Change from “extraction” to “acquisition”
Figure 1: Sentinel-2 MSI 20-m spatial resolution is not appropriate for high accuracy informal settlement mapping as stated in the paper.
Reviewer 3 Report
The research is about a land use classification focusing on identifying informal settlements in a study area in South Africa. The main objective is to test and compare different feature combinations for a machine learning classification (based on the random forest algorithm) within Google Earth Engine. The compared feature sets are composed of spectral bands (SBs), spectral indices (SIs), spectral bands plus spectral indices (SBS + SIs), texture metrics (Txts), spectral bands plus texture metrics (SBs+Txts), texture metrics plus spectral indices (SIs+Txts), and spectral bands+spectral indices+texture metrics (SBs+SIs+Txts) all derived from sentinel2A imagery. The comparisons are both visual and based on a statistical approach. The paper is well written and organized and may interest the RS readers. However, some points should be addressed before considering it for publication:
- In the introduction, other research is considered where GEE was used. Many of them are not so crucial for the specific application of the present study. Differently, other research much more relevant to RF land cover classification in GEE (mainly using textural information from GLCM) are not considered, e.g.:
o Jin, Y.; Liu, X.; Chen, Y.; Liang, X. Land-cover mapping using Random Forest classification and incorporating NDVI time-series and texture: a case study of central Shandong. Int. J. Remote Sens. 2018, doi:10.1080/01431161.2018.1490976.
o Loukika, K.N.; Keesara, V.R.; Sridhar, V. Analysis of Land Use and Land Cover Using Machine Learning Algorithms on Google Earth Engine for Munneru River Basin, India. Sustainability 2021, 13, 13758. https://doi.org/10.3390/su132413758
- Please address this issue.
- The objectives, stated at the end of the introduction, should be verified and improved in terms of overall coherency and conciseness;
- Figure 1 should be improved e.g., using a topographic map in (b) and an image with better resolution in (c)
- Line 248, do you mean the COPERNICUS/S2_SR (surface reflectance) dataset?
- The pre-process developed on S2 images should be better described by indicating and describing all the steps (i.e., cloud masking, time filtering, reducing) in the correct order and coherence with the flow chart.
- Table 1, the number of features is not coherent with the text of line 269;
- Table 2, please add a column containing the main reference for each spectral index;
- GLCM analysis: GEE includes a GLCM algorithm. Why did you decide to develop this analysis in ENVI? This external step increases the complexity and the reproducibility of all the procedure
- Feature combination (lines 331-341): running an RF classification in GEE, including up to 44 bands, could be impossible, even for small study areas. A feature reduction based on linear approaches (e.g. PCA) could have been considered.
- Table 3 is a small flow chart that could be eliminated
- Small polygons as training samples (lines 358-359). How many samples did you use? Were training and validation sample points extracted from the identical polygons?
- lines 373-374: “variable importance scores for all features were calculated using the RF classifier”. Please clarify and add technical information to this step.
- Eq1 seems to be a simply F-score. Why do you use the terms “F1-measures” and “F1 score” in the text? Please improve coherency on this point
- Figure 3, please improve the graphical quality of this map (e.g., removing the title)
- Figure 4: Please explain the meaning of the black square. Otherwise, remove it.
- Table 4: even though it is a subsequent step, you could insert here the output of the statistical tests in table 6. Probably the T-tests could follow the accuracy analysis since it is performed on the results from this step.
- Table 7, please use hectares for the patches’ area. The difference between classified and reference could be indicated as well also in percentage.
- Figure 7 is a repetition of a part of figure 4. Please consider to remove it.
- This kind of land use classification, even in GEE, is moving towards object-based approaches using high-resolution images. Planetscope images will be available also in South Africa. This point should also be discussed considering recent relevant works:
o Rufin, P.; Bey, A.; Picoli, M.; Meyfroidt, ; Patrick An operational framework for large-area mapping of active cropland and short-term 2 fallows in smallholder landscapes using PlanetScope data. Int. J. Appl. Earth Obs. Geoinf. 2022, 112, 102937, doi:10.1016/J.JAG.2022.102937.
o Vizzari, M. PlanetScope, Sentinel-2, and Sentinel-1 Data Integration for Object-Based Land Cover Classification in Google Earth Engine. Remote Sens. 2022, Vol. 14, Page 2628 2022, 14, 2628, doi:10.3390/RS14112628.
- The conclusions are too long. Please reword and synthesize them, focusing on the primary outcomes and importance of the research, lessons learned, and future perspectives.
Reviewer 4 Report
The paper is well done and it is quite original. It argues about very interesting and topical issues. The aim of paper is clear. The figures are clear and well done. Conclusions are justified from the results and discussion. References are updated and necessary. Its significance is high and it could have a good impact in the scientific world.
Only few things to correct:
1) Line 320 and 398: Giannini and Merola (320); Zurqani and Post (398): why put the names of these authors? Earlier, you just inserted the reference as a number. Please insert only the number.
2) line 374: OOB: please, make this acronym explicit first.
3) lines 524-525: there is a formatting problem to be solved.
4) line 667: Estimated informal settlement areas: I think there is a formatting problem to be solved.
Reviewer 5 Report
Google Earth Engine for informal settlement mapping: A Random Forest classification using spectral and textural information
In this manuscript, the authors leverage cloud-based computing techniques within Google Earth to integrate spectral and textural features in order to alleviate complexities in IS extraction. The manuscript aims to investigate the potential and advantages of Google Earth’s innovative image processing techniques for accurate Informal Settlement mapping. The authors investigated seven data input models derived from Sentinel 2A bands, band-derived texture metrics, and spectral indices (NDVI, NDWI, NDBI) through a random forest (RF) supervised protocol. The main objective was to explore the value of different data input combinations in accurately depicting IS locations and extents. Results revealed that the classification based on spectral bands + textural information yielded the highest informal settlement identification accuracy (94% F1- score). The addition of spectral indices decreased mapping accuracy. Spatial accuracy assessment revealed that the textural features’ model achieved the highest spatial accuracy.
This manuscript is well structured, use quite good scientific language and the topic is interesting. After I read this manuscript, I can recommend it for publication after the following suggestions are considered:
§ Try to avoid some of the acronyms in the abstract. I know in this particular case it is not such easy, but you should do a proper effort. At least try to reduce the number as much as you can.
§ In the abstract, the authors should clarify the message and be more concise. Why it is important your research and how you do.
§ Also in the abstract, the authors argue “for accurate Informal Settlement mapping”. Why do they use “Informal Settlement” in capital letters? I would also rephrase this like “for accurate and detailed mapping of informal settlements”.
§ Some sentences are just confusing or reiterative, ie. “Spatial accuracy assessment revealed that the textural features’ model achieved the highest spatial accuracy”. The authors should rephrase the sentence. “Our results confirm that the highest spatial accuracy is achieved with the textural features’ model”
§ The authors start in the introduction with the definition or informal settlements by arguing “Informal settlements have become the new reality in urban landscapes, especially in most cities of the developing world. Sometimes called “favelas”, “shanties”, “ghettos”, “unplanned townships” or “slums” [1], informal settlements are areas inhabited by an increasing number of dwellers living in overcrowded conditions, and often deprived of legal tenure [2], durable housing and basic services [3].” Everything that the authors argue here it is correct, but in my eyes this needs a little bit more context. For example, where, why and how much these informal settlements are expanding? (“Recent statistics have indicated that, of the four billion people who are currently residing in urban areas [4], 1.6 billion live in informal areas [5], a figure that is estimated to rise to 3 billion by the mid-21st century [6].”) Is that a problem from poor or developing cities or everywhere? I would recommend you, for this manuscript, to include just only one paragraph referring to the exceptional importance of recent and fast urbanization, particularly in some developing countries, where it is emerging false urbanization processes (the cities attract more people than they can keep). Also, I would recommend you to give a global perspective to the problem. For example, in the paper entitled “Globalization and the shifting centers of gravity of world's human dynamics” the authors argue that, since 1960, the urbanization process at a global scale is moving to the Global South whereas the global wealth is moving to the Global East. I would suggest you to include this study and other similar in your introduction or discussion.
§ A second aspect related to the former argument is that informal settlements can be also a reality in rich countries. This is particularly true in the context of increasing inequalities. For example, lets think about the impact of refugees in European countries in the last decade, where many of them have the risk to live in guettos or in a parallel society. For example, Bakker et al. (2019) measuring Fine-Grained Multidimensional Integration Using Mobile Phone Metadata for The Case of Syrian Refugees in Turkey. I recommend you also to include a little bit discussion on this point.
§ The authors argue “In line with the 2030 Agenda for Sustainable Development [7], countries are mandated to transform all informal settlements into improved and serviced formal neighborhoods.” I understand that the idyllic forecasts of this report, but they are unrealistic, especially for a short time of period. You could say something more real such “According to 2030 Agenda, it is expected to increase efforts against informal settlements, to reduce the number or for improving the quality of life of their residents” Just for example.
§ The authors argue “city planners and policy makers need information on their location and extent [10], which is often scanty, not up-to-date, or inaccurate [9, 11].” That is a great point to reinforce. Also the problem of these informal settlements is that they expand much faster than authorities realize, because these have not means or technologies or enough staff.
§ I totally agree with the authors with the great advantage of remote sensing and aerial imagery compared to official census. I would suggest to be explicit with the advantages: more updated information, more reliable, less people staff requirements, objective information (not based on subjective criteria like some census or surveys), etc
§ The authors argue: “Of late, VHR imagery has been largely used in the mapping of informal settlement landscapes [2, 11, 18, 19]. However, accurate informal settlement extraction continues to be hindered by (1) heterogeneity and complex spectral characteristics of urban land [20], (2) fragmented spatial configuration of many cities [21] and diversity of morphologies of informal settlements [22]. These characteristics vary extensively between countries and cities, within cities, and socio-economic contexts, making difficult the characterization of data input combinations necessary for precise mapping of these deprived areas [23].” At this point, an important concern. I totally agree with this point and I think it is very crucial for your research. In the past, studies were not aware of this aspect and recently became a more important factor to consider in studies whose methodologies could be globally expanded. For example, some recent studies refer to the “Scale, context, and heterogeneity: the complexity of the social space” for referring to this point. I would suggest you to include a reference to this and any similar research, which make more accurate your argument for social scientists.
§ About the graphical part,
o the authors must include the spatial scale to the small maps shown in Figure 1.
o Table 3 must be improved and clarified. In fact, it is not a table. Looks like more an organigram.
o Figures 3, 5 and 7 should be expanded and the text optimized.
Round 2
Reviewer 3 Report
The Authors successfully addressed all the highlighted issues. However, some minor points should be resolved:
- Figure 1: the relationship between the (a) and the (b) sub-figures is unclear. May you use an arrow to connect them? Moreover, the (a) sub-figure should be mentioned in the caption.
- Figure 2: I think the flowchart is critical for presenting the methodological workflow. In this regard, the overall graphical quality may be improved. In particular, the "Sentinel 2A Image collection" should be connected with the "filtering" item. GEE platform should not be included in a container since it is referred to all the environment. The "Outside" label should be changed with the ENVI one.
- ll 608-611: The suggested reference is a methodological step forward compared to the mentioned one since the use of a 6-month Planet composite (available in GEE through the NCFI program), combined with Sentinel 2 and Sentinel 1 data within an object-based approach, seems to improve the geometrical and thematic accuracy of the output.
